# Evaluating the Use of a Similarity Index (SI) Combined with near Infrared (NIR) Spectroscopy as Method in Meat Species Authenticity

**DOI:** 10.3390/foods12010182

**Published:** 2023-01-01

**Authors:** Daniel Cozzolino, Daniel Bureš, Louwrens C. Hoffman

**Affiliations:** 1Centre for Nutrition and Food Sciences (CNAFS), Queensland Alliance for Agriculture and Food Innovation (QAAFI), The University of Queensland, Brisbane, QLD 4072, Australia; 2Institute of Animal Science, 104 00 Přátelství 815, 104 00 Prague, Czech Republic; 3Department of Food Science, Faculty of Agrobiology, Food and Natural Resources, Czech University of Life Sciences Prague, 165 00 Prague, Czech Republic

**Keywords:** exotic species, similarity index, meat, NIR, chemometrics

## Abstract

A hand-held near infrared (NIR) spectrophotometer combined with a similarity index (SI) method was evaluated to identify meat samples sourced from exotic and traditional meat species. Fresh meat cuts of lamb (*Ovis aries*), emu (*Dromaius novaehollandiae*), camel (*Camelus dromedarius*), and beef (*Bos taurus*) sourced from a commercial abattoir were used and analyzed using a hand-held NIR spectrophotometer. The NIR spectra of the commercial and exotic meat samples were analyzed using principal component analysis (PCA), linear discriminant analysis (LDA), and a similarity index (SI). The overall accuracy of the LDA models was 87.8%. Generally, the results of this study indicated that SI combined with NIR spectroscopy can distinguish meat samples sourced from different animal species. In future, we can expect that methods such as SI will improve the implementation of NIR spectroscopy in the meat and food industries as this method can be rapid, handy, affordable, and easy to understand for users and customers.

## 1. Introduction

Red meat represents a significant proportion of the humans’ daily diet as it provides nutrients such as protein, vitamins, and minerals, which are essential to maintain a healthy life [1,2,3,4,5]. Different sources of red meats are available and used as a supply of protein such as pork, beef, and lamb as well as wild species in some countries [4,5,6,7,8]. With the growing consumption of red meat and meat products, the consumer is more aware of issues associated with meat safety such as authenticity [3,9,10,11].

The most recent issues associated with both meat authenticity and fraud involved the replacement of high-value ingredients with not-expensive ones such as horse (e.g., horse meat scandal) [3,12,13,14,15,16]. In other cases, authenticity is associated with the consumption of certain species proscribed by religious reasons (e.g., pork in Muslim countries). The meat industry is also driven by the need to supply the consumer with a consistent high-quality product at an affordable price [4]. Consequently, these issues have increased awareness about authenticity and fraud in the meat industry [3].

Authentication and the recognition of species have been a major threat for the modern meat industry, as it decreases the quality and safety of the meat products [17,18]. Testing of animal meat species is essential for evaluating quality and safeguards the consumer against fraudulent activities [17,18]. This is also of importance to guarantee integrity throughout the supply and value chains. Different analytical methods are available and used for meat identification and authentication, including manual inspection, chromatographic methods (e.g., chromatography mass spectrometry), electrophoretic separation of proteins, molecular-biology-based methods, electronic noses, and vibrational spectroscopy (e.g., near, mid, and Raman spectroscopy) [17,18,19]. Some of these methods are subjective (e.g., manual inspection), tedious, time consuming, and inconsistent, while other such as molecular biology-based methods (e.g., DNA based techniques, polymerase chain reaction (PCR), real-time PCR, and multiplex PCR), although precise, are slow and expensive [17,18,19]. Despite these issues associated with the use of traditional methods, vibrational spectroscopy is still considered an emerging technology, which has been proved to be a dynamic and developing method in evaluating and monitoring the authenticity of animal species.

One of the main drawbacks on the utilization of NIR spectroscopy by the food and meat industry is the need of chemometrics to analyze the data collected to make meaningful decisions about the quality and safety of the meat. Chemometrics techniques such as principal component analysis (PCA), discriminant analysis (DA), soft independent modelling of class analogies (SIMCA), and artificial neural networks (ANN) are commonly used to unravel and interpret the spectral properties of the sample, allowing for the classification of samples without the use of direct chemical compositional information [19]. These chemometric techniques have been shown to be able to classify foods, including meat, based on spectral data. However, these advanced chemometrics methods can be difficult to understand and to apply under industrial conditions.

Unlike chemometrics, other qualitative methods, particularly those based on similarity, can be applied to analyze NIR data using “spectral similarity” techniques [20,21,22]. A simple approach for comparing two spectra is the so-called “similarity index” (SI) method, as described by different authors [20,21,22]. The SI method has been used and described to identify pure chemicals (e.g., sugar solutions) [20], to compare and authenticate wines [21], as well as to analyze tobacco leaves [22]. The SI method is created using the measurements of the absorbance for every wavelength of the first spectrum, defined as X variable, where the second spectrum is defined as Y variable. The correlation coefficient (r) is used to compute a similarity index which can be used to test for identity between the samples. In this study, NIR spectra are obtained from a meat sample from a given animal species, then a second meat sample from the same or different animal species is taken, and then the two are correlated to confirm or not the authenticity of the meat sample.

This paper details the application of a similarity index (SI) combined with the near infrared (NIR) spectra of meat samples collected using a hand-held spectrophotometer as a rapid, inexpensive tool to authenticate meat samples sourced from traditional and wild meat species.

## 2. Materials and Methods

Samples of lamb (*Ovis aries*), emu (*Dromaius novaehollandiae*), camel (*Camelus dromedarius*), and beef (*Bos taurus*) were obtained from chilled carcasses after 24 h slaughter and sourced from a commercial slaughterhouse (Queensland, Australia). The fresh meat samples were first cut in small pieces with a knife, and thoroughly hand mixed before being minced. Then, samples were minced using a Tabletop mincer (MEFE 360MC120, 18,000 rpm) fitted with a round mincer plat with 4 mm diameter holes (Mitchell Engineering Food Equipment, Clontorf, Queensland, Australia) which was washed and dried between samples. Four replicates for each species were created (4 animal species × 4 biological replicates = 16).

The near infrared spectra of the minced meat samples were collected using a hand-held NIR spectrophotometer (Micro-NIR 1700. Viavi, Milpitas, CA, USA) operating in the wavelength range between 950 and 1600 nm (10 nm wavelength resolution). The spectra collection and instrument set up were controlled using the proprietary software provided by the instrument manufacturer (MicroNIR Prov 3.1, Viavi, Milpitas, CA, USA). The spectral data acquisition settings were set at 50 ms integration time and an averaging of 50 scans (MicroNIR Prov 3.1, Viavi, Milpitas, CA, USA). Between samples, a reference spectrum was collected using Spectralon^®^. The total number of samples used/scanned was 96 (4 animal species × 4 biological replicates × 6 scans).

The NIR data were exported into The Unscrambler (version X, CAMO, Oslo, Norway) for data analysis and preprocessing. The NIR spectra was preprocessed using the Savitzky-Golay second derivative (21 smoothing points and second polynomial order) prior to spectra interpretation and chemometric analysis [23]. Principal component analysis (PCA) was used to analyze the data and to evaluate the differences or trends associated with the animal species analyzed. The PCA model was developed and validated using full cross validation (leave one out) [24,25,26]. Linear discriminant analysis (LDA) was also used to classify meat samples according to the animal species.

In this study, the Similarity Index (SI) method was used to identify and authenticate the meat species analyzed. The SI is specifically targeted to applications whereby only two spectra are being compared (e.g., beef1 vs. beef2). In this index, the r^2^ is calculated as the coefficient of determination between the absorbance values from the two spectra at each wavelength across the entire wavelength range. This can be easily determined by use of the correlation function in Excel. The inverse relationship with r^2^ means that SI is very sensitive to small changes in r^2^, and SI can range in values from 1 (totally different spectra) to infinity ∞ (identical spectra) [20].
SI = 1/(1 − r^2^)

## 3. Results and Discussion

As the first step, we have interpreted the main features of the NIR spectra of the meat samples analyzed. Figure 1 shows the average of the second derivative NIR spectra of the meat species (e.g., beef, camel, emu, and lamb) analyzed using a hand-held instrument. The second derivative of the NIR spectra of the meat samples showed bands around 985 nm associated with the O-H overtones of water, at 1180 nm (C-H and C=O), at 1205 nm corresponding to a stretching–bending, second overtone of C-H bonds related to lipids [6,9,27]. Additionally, a shoulder around 1350 nm and at 1428 nm O-H stretch first overtone, an O-H combination, and an O-H bend second overtone were noted. These three bands are mainly associated with water content [6,9,27].

A PCA was also performed to observe any trends in the NIR spectra associated with the different meat animal species analyzed. The PCA score plot derived from the analysis of meat samples is shown in Figure 2. The first four principal components (PC) explained 99% of the total variability in the NIR spectra of the meat samples analyzed (PC1 66%, PC2 25%, PC3 5%, and PC4 3%). A separation between meat samples according to the animal species was observed when PC2 vs. PC4 were plotted (Figure 2). The PCA loadings for the first and fourth principal components are reported in Figure 3. The highest loadings in PC2 were observed around 1087 nm (O-H), 1217 nm and 1297 nm (C-H), and at 1428 nm (O-H), while the highest loadings in PC4 were observed at 1050 nm, around 1210 nm (C-H), and 1360 nm (C-H), associated with the presence of lipids (e.g., fatty acid profile) and protein content [6,9,27]. The use of PCA allowed for the identification of differences in the NIR spectra of the meat samples according to the animal species analyzed. These results showed that there is relevant information (e.g., chemical properties) in the NIR spectra that can be used to separate the different animal meat species analyzed.

In addition to the PCA, the NIR spectra of the meat samples were analyzed using linear discriminant analyses (LDA). The second derivative described in the materials and methods was used as a preprocessing method before LDA. The LDA (using 9 latent variables) confusion matrix obtained from the analysis of the meat samples is shown in Table 1. The overall accuracy of the models was 87.8%. It was observed that 92%, 89%, 86%, and 84% of the camel, emu, beef, and lamb meat samples, respectively, analyzed using NIR spectroscopy were correctly classified.

After both PCA and LDA analysis, the similarity index (SI) was calculated. As defined in the previous section, similar samples will have a correlation very close to one (r = 1.0). The SI calculated in this study was considered a more sensitive measure of similarity in comparison with other methods as reported by other authors [20]. In this way, an SI will differ between 1.0 for totally different spectra (e.g., different meat species) and infinity for identical species. In this study, the SI calculated according to previous reports [20,21] was chosen as the indicator of similarity between the same species of meat (e.g., beef1 vs beef 2) [20,21]. The results of the SI for the comparison of the meat samples analyzed using the whole NIR spectra (950 to 1600 nm) are shown in Figure 4, Panel A. The results showed that an SI value > 350 corresponds to a r^2^ value > 0.997. This result was considered adequate to either identify the different meat species or the similar ones. Therefore, this value was set as the minimum value for similarity, meaning that the meat samples from the same animal species will have at least a value equal or higher than 350, where meat samples having values below this limit were considered different. It was also observed that the SI successfully matched all identical meat samples corresponding to the same species (e.g., beef vs. beef; lamb vs. lamb, etc.).

Figure 4, Panel B shows the results for the SI calculated using the NIR range between 1240 and 1400 nm associated with the C-H bonds, related to the range corresponding to lipid and protein contents [6,9,26]. The trends observed were similar to those described in Figure 4, although the threshold has changed to SI > 750 in most of the samples and mixtures analyzed. However, for the meat samples obtained from lamb, the threshold was lower. This can be also due to differences in fat content. Overall, it can be stated that if a sample has a similarity index over 350, the samples can be considered as identical. In instances where the spectra are to a lesser extent easily separated, further research is needed to determine what number is best to be used as the SI.

The utilization of the similarity index method applied to the NIR spectra of meat samples has been proven to be an alternative tool to other classification methods to distinguish samples from the same animal species from different ones (e.g., traditional vs. wild meat). However, research into the overall use of the SI method should also be extended to observe the effects of mixtures, breeds, etc., as it seems the SI is dependent on the data set and experimental conditions. As described above, this simple approach (SI) for comparing two spectra has been described by other authors using different food matrices [20,21,22]. Overall, the results from this study were comparable with those studies that have analyzed liquid samples such as wine, tobacco leaves, and sugars [20,21,22]. In addition, the results obtained in this study from the application of the SI are comparable to those obtained using classical chemometrics methods such as LDA (Table 1). The main advantage of using SI over classical chemometric methods (e.g., LDA, PCA) is that this index can be easily understood by the nonexpert where various operators in the industry with a diverse skill base can use this method to trace the origin of the meat. In addition, an SI system must be able to be integrated and operated using readily available equipment (e.g., portable NIR instruments). Ultimately, the use of an SI can be inexpensive and can be implemented on commonly used software such as Excel^®^.

## 4. Conclusions

In this study, the use of an SI method to compare the NIR spectra of different animal species was evaluated. The SI method has shown that it can be used as an alternative to other classification methods available such as linear discriminant analysis. Overall, these results indicate that SI combined with NIR spectroscopy can distinguish meat samples sourced from different animal species (e.g., traditional vs. wild meat species). In future, we can expect that methods such as SI will improve the implementation of NIR spectroscopy in the meat and food industries as an authentication tool that is quick, handy, and affordable for customers.

## Figures and Tables

**Figure 1 foods-12-00182-f001:**
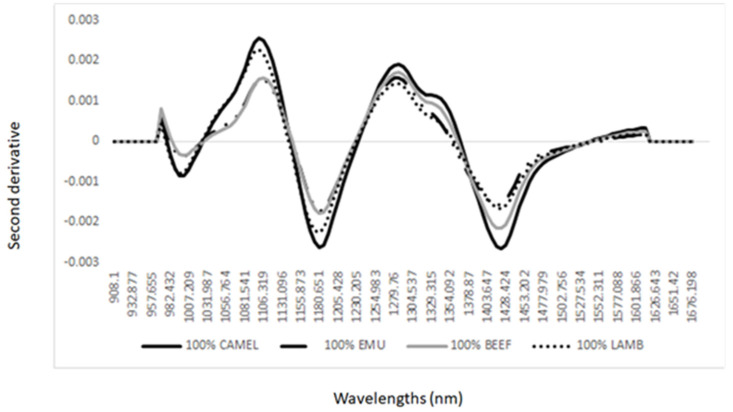
Average second derivative NIR spectra of lamb (*Ovis aries*), emu (*Dromaius novaehollandiae*), camel (*Camelus dromedarius*), and beef (*Bos taurus*) minced samples analyzed using a portable NIR instrument.

**Figure 2 foods-12-00182-f002:**
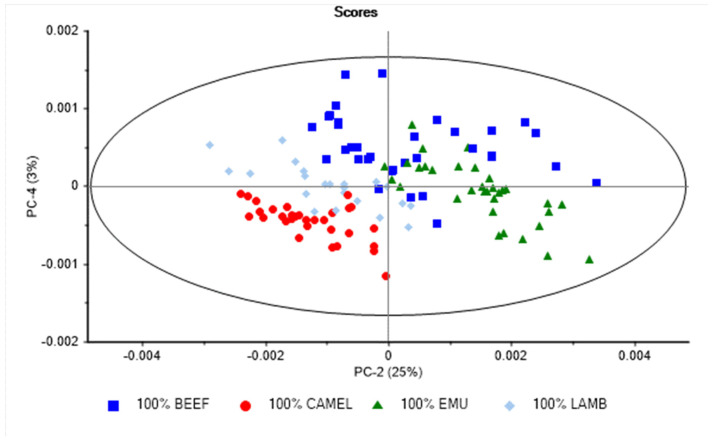
Principal component score plot of NIR spectra of lamb (*Ovis aries*), emu (*Dromaius novaehollandiae*), camel (*Camelus dromedarius*), and beef (*Bos taurus*) minced samples analyzed using a portable NIR instrument.

**Figure 3 foods-12-00182-f003:**
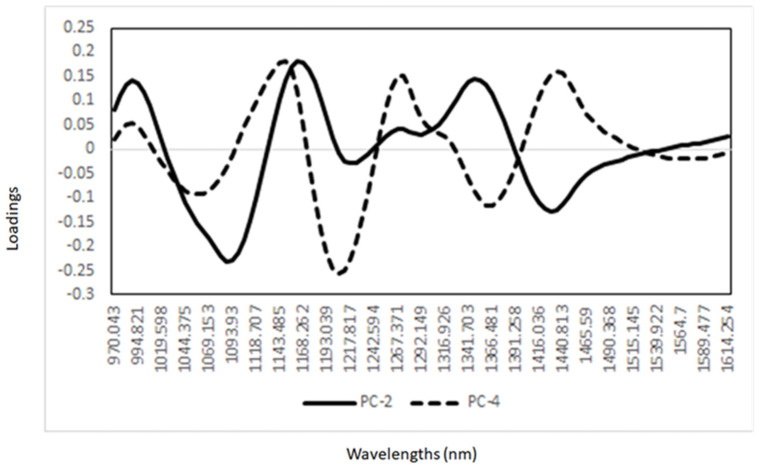
Principal component loadings derived from the analysis of minced meat samples.

**Figure 4 foods-12-00182-f004:**
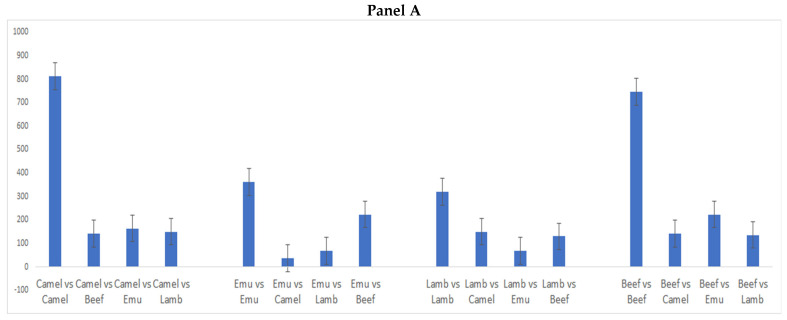
Similarity index calculated for the minced meat samples of lamb (*Ovis aries*), emu (*Dromaius novaehollandiae*), camel (*Camelus dromedarius*), and beef (*Bos taurus*) analyzed using a portable NIR instrument. (**Panel A**): whole range (950 to 1600 nm); (**Panel B**): lipid and protein range (1200 to 1400 nm).

**Table 1 foods-12-00182-t001:** Linear discriminant analysis confusion matrix of meat samples analyzed using near infrared reflectance spectroscopy. In brackets are the percentages of correct classification.

	Camel	Emu	Beef	Lamb
Camel	34 (92%)	0	0	3
Emu	0	33 (89%)	3	0
Beef	4	0	32 (86%)	1
Lamb	2	0	3	31 (84%)

## Data Availability

The data are available from the corresponding author.

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
