# Peer review of "Evaluating the Use of a Similarity Index (SI) Combined with near Infrared (NIR) Spectroscopy as Method in Meat Species Authenticity"

_foods, 2023, doi:10.3390/foods12010182_

Round 1

Reviewer 1 Report

The paper describes the use of a new parameter (similarity index), applied to NIR spectra for characterization of different types of meats. The method shows good results for authenticity of meat with simple methodology. 

Tittle: Evaluating the use of a similarity index (SI) as method in meat species authenticity. 

Manuscript ID: foods-2069014

Title

I consider that the title should include the use of NIRS to show where the SI is applied. 

Materials and methods

Line 93. In this line, you indicate that you obtain 50 scans per samples, however, in the line 95, you say 6 scans. Please, to clarify this. Anyway, 6 scans of the six samples should not be considered as independent samples. In that case, you should use, segmented cross-validation, leaving the six scans on the same sample out each time. 

Results and discussion

Figure 2 should be colored by type of meat to be more illustrative or include the mean spectra for each type of meat, in order to see the visual differences. 

Line 132. In this sentence, when you say PC1 vs PC4… it should be… when the scores for PC1 vs the scores for PC4. On the other hand, you write that the separation is corresponding to these two components, however, in the Figure 2, it doesn’t appear PC1. Please, correct this mistake. 

Line 134. Could you describe which are 1180 and 1428 nm bands associated?

Line 146. You should indicate how many latent variables are used in LDA and if the spectra were normalized or not. 

Line 160. It is not clear how Figure 4 is obtained. Is it the mean of SI between samples? Which means the deviation standard bar? Try to rewrite this information to better understanding of readers. Also, how the threshold for the SI is calculated. How do you calculate that SI = 350 correspond with a r2 = 0.9999? If I use the equation at the line 112 I obtain a different value. 

Figure 4, Panel B. Why does the deviation standard not appear?

General comment

Did you try to build models based on the SI? 

I consider that some validation samples should be included. 

Author Response

REV 1

We thank the reviewer for the comments and suggestions to improve the clarity of our manuscript.  Our answers are below in bold.

  • Title

I consider that the title should include the use of NIRS to show where the SI is applied.  We have changed the title as suggested by the reviewer to “Evaluating the use of a similarity index (SI) combined with near infrared (NIR) spectroscopy as method in meat species authenticity”.

  • Materials and methods

Line 93. In this line, you indicate that you obtain 50 scans per samples, however, in the line 95, you say 6 scans. Please, to clarify this. Anyway, 6 scans of the six samples should not be considered as independent samples. In that case, you should use, segmented cross-validation, leaving the six scans on the same sample out each time.  The 50 refers to the internal scanning during the spectra collection, this refers to the average scans collected. This is a routine protocol used in most of the NIR instruments (e.g. handheld, lab base, etc). Then for each sample we have then collected 6 scans from different positions.

  • Results and discussion

Figure 2 should be colored by type of meat to be more illustrative or include the mean spectra for each type of meat, in order to see the visual differences.  We think that the author refers to Figure 1 and not to figure 2.  We have added the mean spectra for each of the type meat samples as suggested by the reviewer. 

Line 132. In this sentence, when you say PC1 vs PC4… it should be… when the scores for PC1 vs the scores for PC4. On the other hand, you write that the separation is corresponding to these two components, however, in the Figure 2, it doesn’t appear PC1. Please, correct this mistake. Sorry, this was a typo. We have corrected as suggested by the reviewer.

Line 134. Could you describe which are 1180 and 1428 nm bands associated? We have added the information as suggested by the reviewer.

Line 146. You should indicate how many latent variables are used in LDA and if the spectra were normalized or not. We have added this information in the text as suggested by the reviewer.

Line 160. It is not clear how Figure 4 is obtained. Is it the mean of SI between samples? Which means the deviation standard bar? Try to rewrite this information to better understanding of readers. Also, how the threshold for the SI is calculated. How do you calculate that SI = 350 correspond with a r2 = 0.9999? If I use the equation at the line 112 I obtain a different value.   We have reworded the section. We are surprises from the reviewer comments as the whole NIR spectra should be used to calculate the SI.  We have used the whole NIR spectra, the whole series of absorbances at each wavelength corresponding to Sample A is regressed against the whole spectra (absorbances) of sample B.  This is how the SI is calculated. We are not compared the absorbance at a single wavelength.

Figure 4, Panel B. Why does the deviation standard not appear? This is the value of the SI.  Please refer to the Materials and Methods section. An index does not have a SD.

  • General comment

Did you try to build models based on the SI? No, we did not. The SI is the correlation between the whole NIR spectra for each sample.  The SI cannot be used to develop a calibration model.

I consider that some validation samples should be included.  No validation is needed for SI. I will suggest the reviewer to read the original paper published by Tony Davies and collaborators.  The validation that we have used in this paper was to select different regions and demonstrate that the SI do the job.

Reviewer 2 Report

Manuscript Number: FOODS-2069014

Decision Letter of Article: Evaluating the use of a similarity index (SI) as method in meat species authenticity

This paper showed an interesting study about development of a new method to assess the meat authenticity using the similarity index. Meanwhile, it was raising some questions that must be explained by the authors. 

1-The authors performed another biochemical test such as PCR, or other technique for validate the samples? There is any correlation about both data?

2-The cells of blood can interfere negative with the FTIR analysis. How did the authors overtake or optimize this situation?

3-The figures must be improved in terms of letter size and resolution.

4-The sample size is quite small. How can they justify the results based on these experimental data?

5-Why the authors did not use other classification SIMCA, KNN or and DA algorithms for classification models? Which is the real advantage of those methodologies in this study compared with other models then created?

The work must be reformulated and improved due to the failures and doubts. For those reasons, I suggest a major revision. The questions that were raised must be carefully answered.

Author Response

REV 2

We thank the reviewer for the comments and suggestions to improve the clarity of our manuscript.  Our answers are below in bold.

1-The authors performed another biochemical test such as PCR, or other technique for validate the samples? There is any correlation about both data?  We did not use any other biochemical methods as suggested by the reviewer to identify the species as we collected the samples from the animals and were present during the slaughter of the animals.  We have used and applied the same methodology for the last 30 years.  In addition, most of the referenced papers related with authentication use similar protocols. 

2-The cells of blood can interfere negative with the FTIR analysis. How did the authors overtake or optimize this situation? We have used NIR and NOT FTIR as suggested by the reviewer.  We have used and applied NIR spectroscopy combined with chemometrics to analyse meat for the last 30 years and we did not encounter this issue.

3-The figures must be improved in terms of letter size and resolution. We have improved the figures (size) as suggested by the reviewer.

4-The sample size is quite small. How can they justify the results based on these experimental data? The use of SI is not dependent of the number of samples.  The reviewer can find more information on the technique on the references used.  This is one of the many advantages of using SI compared with chemometrics.

5-Why the authors did not use other classification SIMCA, KNN or and DA algorithms for classification models? Which is the real advantage of those methodologies in this study compared with other models then created?  We have explained this in the Introduction.  We have added the text here to highlight to the reviewer the main objective of this study.  Chemometrics techniques such as principal component analysis (PCA), discriminant analysis (DA), soft independent modelling of class analogies (SIMCA), and artificial neural networks (ANN) are commonly used to unravel and interpret the spectral properties of the sample, allowing for the classification of samples without the use of direct chemical compositional information [19]. These chemometric techniques have been shown to be able to classify foods, including meat based on spectral data.  However, these advanced chemometrics methods can be difficult to understand and to apply under industrial conditions.

Unlike chemometrics quantitative or qualitative methods, similarity can be characterised using spectral similarity [20-22].  A simple approach for comparing two spectra is the so-called “similarity index” (SI) method, as described by different authors [20-22].  The SI method has been used and described to identify pure chemicals (e.g. sugars) [20], to compare and authenticate wines [21] and applied to analyse tobacco leaves [22].  The SI method is created using the measurements of the absorbance for every wavelength of the first spectrum, defined as X variable, where the second spectra is defined as Y variable.  The correlation coefficient (r) is used to compute a similarity index which can be used to test for identity between the samples.  In this study a NIR spectra is obtained from a meat sample from a given animal species then a second meat sample from the same animal species is taken, then the two are correlated to confirm the authenticity of the meat sample.

Round 2

Reviewer 1 Report

Thank you for all the corrections. 

I consider most of the comments were well answered. However, there are two relevant points that still require corrections. 

Line 167.

This part was not corrected. The sentence “The results showed that a SI value > 350, corresponds to a r2 value of 0.999999.” is not correct. 

When you calculate the r2 for an SI = 350, applying the equation: , the value that you obtain is 0.997.

350 = 1/ (1-r2)

350 – 350r2 = 1

349 = 350r2

r= 0.997

Consequently, the sentence should be:

The results showed that a SI value > 350 corresponds to a r2 value > 0.997. This result is still a good result but correct.  

Figure 4.

You are describing two panels: A and B. However, you don’t describe both panels in the Figure caption 4. I understood that panel A is for the whole range and the panel B for the region related to lipid and protein contents, but this is not described. 

On the other hand, you include bar errors in panel A, but not in panel B. Please, do it also in panel B. 

Finally, in line 186, you refer to Figure 3 and it should be Figure 4 (panel A). 

Author Response

Rev 1.

  • Line 167. This part was not corrected. The sentence “The results showed that a SI value > 350, corresponds to a r2 value of 0.999999.” is not correct.   When you calculate the r2 for an SI = 350, applying the equation: , the value that you obtain is 0.997.

350 = 1/ (1-r2)

350 – 350r2 = 1

349 = 350r2

r2 = 0.997 

Consequently, the sentence should be:

The results showed that a SI value > 350 corresponds to a r2 value > 0.997. This result is still a good result but correct.  Thank you for the suggestion. We have corrected as suggested by the reviewer.

  • Figure 4. You are describing two panels: A and B. However, you don’t describe both panels in the Figure caption 4. I understood that panel A is for the whole range and the panel B for the region related to lipid and protein contents, but this is not described. We have added the description in the figure caption as suggested by the reviewer.
  • On the other hand, you include bar errors in panel A, but not in panel B. Please, do it also in panel B. We have corrected and added the error bars as suggested by the reviewer.
  • Finally, in line 186, you refer to Figure 3 and it should be Figure 4 (panel A). We have corrected as suggested by the reviewer.

Reviewer 2 Report

Manuscript Number: FOODS-2069014

Decision Letter of Article: Evaluating the use of a similarity index (SI) as method in meat species authenticity 

This paper showed an interesting study about development of a new method to assess the meat authenticity using the similarity index. Meanwhile, it was raising some questions that must be explained by the authors. 

1-The authors performed another biochemical test such as PCR, or other technique for validate the samples? There is any correlation about both data?

2-The cells of blood can interfere negative with the FTIR analysis. How did the authors overtake or optimize this situation?

3-The figures must be improved in terms of letter size and resolution.

4-The sample size is quite small. How can they justify the results based on these experimental data?

5-Why the authors did not use other classification SIMCA, KNN or and DA algorithms for classification models? Which is the real advantage of those methodologies in this study comparatively with other models then created?

The work must be reformulated and improved due to the fails and doubts. For those reasons, I suggest a major revision. The questions that were raised must be carefully answered.

Author Response

Rev 2.

It is very disappointing that the reviewer is not aware of the vast amount of research in the field of NIR spectroscopy apply to meat sciences (e.g. quality, authenticity, species identification, etc).  In addition, we have already answered the questions and explained to the reviewer these issues in our previous version. 

We like to clarify to the reviewer that we did not use FTIR.  The reviewer is still confused, wrong and not familiar with the technique that we have used and reported in this research. We are using diffuse reflectance spectroscopy and NOT FTIR as the reviewer continue to point out.  Our previous research as well as other researchers in the field have proved that did has little interference with the NIR spectra.  We already have optimised the sample presentation based on our research.

 The reviewer should be aware that most of the papers related with identification and authentication of meat are based on the use of authentic samples.  As explained before, we have collected samples from our experimental research station where we know the identity of the samples.  We do not need to do a PCR to test this fact.  In addition, the use of unique samples is a common practice in the application of NIR not only in meat but also in other applications dealing with authenticity and identification. The reviewer can be enlightened and learn by reading recent reviews in the field published by in this journal, other scientific journals and books. 

All figures were improved. 

We have already explained in the Introduction the main objective of this paper.  We are not comparing the use of similarity index with other chemometrics techniques.  We are presenting and reporting the use of an easy-to-use method to identify meat samples. 

The reviewer also is not aware or ignore how the use of simple statistics such as the similarity index are used in practice.  For example, the pharmaceutical industry has used similarity and conformation indexes for years.  The implementation of an index is a straightforward method that do not require the knowledge of chemometrics and it is easy to implement and understand. 

1-The authors performed another biochemical test such as PCR, or other technique for validate the samples? There is any correlation about both data?

2-The cells of blood can interfere negative with the FTIR analysis. How did the authors overtake or optimize this situation?

3-The figures must be improved in terms of letter size and resolution.

4-The sample size is quite small. How can they justify the results based on these experimental data?

5-Why the authors did not use other classification SIMCA, KNN or and DA algorithms for classification models? Which is the real advantage of those methodologies in this study comparatively with other models then created?

The work must be reformulated and improved due to the fails and doubts. For those reasons, I suggest a major revision. The questions that were raised must be carefully answered.